# *ESR1* Methylation Measured in Cell-Free DNA to Evaluate Endocrine Resistance in Metastatic Breast Cancer Patients

**DOI:** 10.3390/ijms23105631

**Published:** 2022-05-18

**Authors:** Manouk K. Bos, Teoman Deger, Stefan Sleijfer, John W. M. Martens, Saskia M. Wilting

**Affiliations:** Department of Medical Oncology, Erasmus MC Cancer Institute, Erasmus University Medical Center, Dr. Molewaterplein 40, P.O. Box 2040, 3000 CA Rotterdam, The Netherlands; t.deger@erasmusmc.nl (T.D.); s.sleijfer@erasmusmc.nl (S.S.); j.martens@erasmusmc.nl (J.W.M.M.)

**Keywords:** *ESR1* methylation, circulating tumor DNA, metastatic breast cancer

## Abstract

*ESR1* methylation was proposed as mechanism for endocrine resistance in metastatic breast cancer patients. To evaluate its potential as a minimally invasive biomarker, we investigated the feasibility of measuring *ESR1* methylation in cell-free DNA (cfDNA) and its association with endocrine resistance. First, we provided evidence that demethylation in vitro restores ER expression. Subsequently, we found that *ESR1* methylation in cfDNA was not enriched in endocrine-resistant versus endocrine-sensitive patients. Interestingly, we found a correlation between *ESR1* methylation and age. Publicly available data confirm an age-related increase in *ESR1* methylation in leukocytes, confounding the determination of the *ESR1* methylation status of tumors using cfDNA.

## 1. Introduction

Estrogen is a hormone that exerts a variety of functions in both reproductive tissues and nonreproductive tissues through the activation of ligand-dependent transcription factors, one of which is ERα, encoded by the estrogen receptor 1 gene (*ESR1*). ERα-mediated transcription is regulated at different levels and involves both direct and indirect mechanisms. ER can directly bind to specific DNA sequences called estrogen response elements (EREs), but ERs can also be recruited onto DNA by tethering via interactions with other transcription factors that regulate transcription in an ERE-independent manner. ERα-mediated transcription is therefore a complex process that involves various coregulatory proteins and transcription factors [1,2].

In addition to its physiological role, ERα is also associated with the development and progression of breast cancer. In patients with estrogen receptor (ER)-positive breast cancer, endocrine therapy, which works by either blocking estrogens from attaching to the ER or by deprivation of estrogen, remains the mainstay of treatment. Although most patients initially respond to endocrine therapy, ultimately all patients will develop endocrine resistance. This poses an important challenge in patients with ER-positive breast cancer, and biomarkers are needed to guide clinical management. Different mechanisms underlying endocrine resistance were identified in metastatic breast cancer (MBC) patients, one of the most recognized mechanisms being the occurrence of activating hotspot mutations in *ESR1*, *ESR1* hotspot mutations are found in around 40% of patients with endocrine-resistant breast cancer [3,4,5,6,7]. They are located within the ligand-binding domain of the gene and result in an estrogen-independent constitutive activation of the ER [8,9]. This enables breast cancer cells to grow independently of the presence of estrogen.

Next to mutations in *ESR1*, other mechanisms have been proposed to induce estrogen-independent growth of breast cancer cells, highlighting the complex gene regulation of *ESR1*. For example, inactivating mutations in neurofibromin (*NF1)*, which functions as an ER transcriptional co-repressor [10], were found in endocrine-resistant breast cancer [7]. Over the past few years, the loss of the ER by epigenetic silencing of *ESR1* has received increased attention as a potential alternative mechanism of endocrine resistance. Several studies found a correlation between the hypermethylation of cytosine guanine phosphodiesters (CpG dinucleotides) of the *ESR1* promoter and decreased ER expression in breast cancer tissue [11,12,13,14,15,16], and few studies also found a correlation of *ESR1* promoter methylation with outcomes of endocrine treatment [11,17]. However, whether there is a direct causal association between *ESR1* methylation and ER expression has been both proposed [16] and disputed [18].

Following the hypothesis that *ESR1* methylation is associated with acquired resistance to endocrine treatment, it is expected to arise during treatment. To test this hypothesis, studying *ESR1* methylation repetitively during treatment is required. A non-invasive tool to study tumor biology in blood is cell-free DNA (cfDNA), which is fragmented DNA present in the circulation likely composed of all dying cells in the body. As such, cfDNA mostly originates from leukocytes [19]. However, in patients with cancer, a varying fraction of the DNA that circulates in blood is tumor-derived (ctDNA). In most patients with metastatic breast cancer, this tumor-specific fraction does not exceed 10% [20]. However, this fraction can be effectively used to repeatedly study the genetic composition of the tumor for purposes such as the monitoring of treatment response [21,22].

The current study aims to evaluate whether determining *ESR1* methylation in cfDNA may provide a minimally invasive biomarker for endocrine resistance. For this purpose, we first established that ER expression was indeed directly associated with DNA methylation in vitro. Subsequently, we evaluated whether the detection of *ESR1* methylation in cfDNA was feasible and associated with endocrine resistance.

## 2. Results

### 2.1. ER Expression Is Regulated by Methylation

We first set out to provide direct evidence that demethylation by 5-aza-2′-deoxycytidine (DAC) restores ER expression in the ER-negative MDA-M-436 cell line. MDA-MB-436 cells were treated with DAC for 5 days. After 5 days, we observed a decrease in *ESR1* methylation, compared to cells without any treatment or treated only with the DAC solvent dimethyl sulfoxide (DMSO, Figure 1 top). Concurringly, the mRNA expression of *ESR1* was restored in these DAC-treated MDA-MB-436 cells compared to untreated and mock-treated cells (Figure 1 bottom). These results show that ER expression is regulated by DNA-methylation, although the exact methylation site responsible for silencing of *ESR1* remains to be elucidated.

### 2.2. ESR1 Methylation Prevalence in cfDNA and Its Prognostic Value

The methylation of *ESR1* was successfully determined in all 31 patients from cohort 1 and 18/20 healthy blood donors (HBDs; EDTA tubes), as well as 17/18 patients from cohort 2 and 7/10 HBDs (CellSave tubes). Figure 2 summarizes the cohorts and samples included in this study. *ESR1* methylation levels of HBDs were not significantly different between Ethylenediaminetetraacetic acid (EDTA, *n* = 7) and CellSave tubes (*n* = 18) from different donors (Mann–Whitney U *p* = 0.809). The median age of patients was higher than the median age of healthy donors in cohort 1 (Mann–Whitney U *p* = 0.009) but not cohort 2 (Mann–Whitney U *p* = 0.391, Appendix A). In patient samples, *ESR1* methylation levels exceeded the cut-off in 12/31 (39%) endocrine treatment-naïve patients from cohort 1 and in 7/17 (41%) patients from cohort 2 post endocrine treatment, which was not significantly different (Fisher’s exact *p* = 1, Figure 3A). Importantly, the presence of *ESR1* methylation in plasma at baseline was not predictive for progression-free survival on first-line treatment with an aromatase inhibitor (AI) in cohort 1 (logrank *p* = 0.660). From the 19 patients in which we detected *ESR1* methylation, 3 also carried an *ESR1* mutation (16%), whereas in the 28 patients without *ESR1* methylation, 8 showed an *ESR1* mutation (29%, Fisher’s exact *p* = 0.319). The proportion of patients who were positive for *ESR1* methylation significantly increased with age (Figure 3B, chi-square test for trend *p* < 0.001).

### 2.3. ESR1 Methylation and Age

Following up on these results in our relatively small cohort, we investigated whether methylation levels in white blood cells were age-related in public data [23]. Interestingly, we found that the methylation of the CpG dinucleotides covered by our assay was moderately, but significantly, associated with age (Figure 3C, Spearman rho = 0.171, *p* < 0.001). This association with age was seen for all CpG dinucleotides in the region of promoter A (Figure 3D) and promoter B (Figure 3E), although the latter association was only weak. CpG dinucleotides in the region of promoter C were highly methylated in white blood cells in all cases, irrespective of age (Figure 3F). The locations of *ESR1* promoter regions and cg probes are summarized in Appendix A.

## 3. Discussion

The primary aim of this study was to evaluate whether *ESR1* methylation was indeed involved in regulation of ER expression and, as such, may provide a minimally invasive biomarker for the detection of endocrine resistance.

Treatment with a demethylating agent increased the expression of *ESR1* in vitro. Although this experiment does not pinpoint which methylation site is responsible for *ESR1* silencing, these findings are in line with other studies suggesting that the methylation of *ESR1* negatively regulates ER expression in vivo [11,12,13,14,15,16]. Notwithstanding the fact that *ESR1* methylation appears to be at least partly involved in the regulation of its expression, we did not find an association between the presence of *ESR1* methylation in cfDNA and progression in endocrine treatment, nor was baseline *ESR1* methylation predictive for outcomes of first-line treatment with AIs. Despite its limitations in sample size, our study was sufficiently powered to detect a similar increase in *ESR1* methylation from endocrine-sensitive to endocrine-resistant patients as was previously observed for *ESR1* mutation incidence. Additionally, the anticipated mutual exclusivity between methylation and mutation was not observed. Instead, we found that the proportion of *ESR1*-methylation-positive patients increased with age. Although this observation was only modest in our patient samples, we found a more pronounced positive correlation with age in WBCs using public data. Irrespective of whether *ESR1* methylation indeed suppresses ER expression, this observation limits the use of *ESR1* methylation in cfDNA as a biomarker of endocrine resistance. None of the earlier studies that investigated *ESR1* methylation in serum or plasma reported an association with age, nor did they describe whether HBDs were age-matched [11,14,24,25]. In our analyses, the reference samples were not fully age-matched as the blood bank from which the reference samples were obtained set the maximum age for blood donation at 80 years. This might have resulted in a slight overestimation of *ESR1* methylation incidence in part of our samples. Based on a mutation analysis, 53% of the patients had detectable ctDNA with a median variant allele frequency of 5.6% anda range from 0.04 to 45%. These were not correlated with the *ESR1* methylation level (Spearman’s rho = −0.066, *p* = 0.719). Together with the fact that we observed a significant correlation between age and *ESR1* promoter methylation in WBCs, it is likely that part of the *ESR1* methylation signal in cfDNA was not tumor-specific. Although we set a cut-off using data from HBDs, and all HBDs were aged >50, our HBDs were not fully age-matched. This observation underlines the importance of using age-matched HBDs when performing epi-genetic analysis of cfDNA, which is often a complicating factor.

The fact that *ESR1* contains multiple promoter regions further complicates formal comparison of findings between studies. Most previous studies focused on promoter A, which is closest to exon 1 and located within a CpG island (i.e., region enriched for CpG dinucleotides). Alternatively, methylation of CpG dinucleotides located in promoter C, and not within promoter A itself, was suggested to be associated with the altered transcription of the ERα gene in AI-resistant cell lines [12]. However, we found that promoter C was highly methylated in WBCs, complicating the analysis of this region in cfDNA. Further underlining the importance of genomic location, another study suggested that the methylation of ER enhancers and not ER promoters were responsible for reduced *ESR1* binding and the decreased expression of ER regulators [18]. Of those ER enhancers, one enhancer site was located within *ESR1* itself. However, this CpG dinucleotide site is also a common SNP, limiting its utility as a routine biomarker when analyzed as a single-marker.

In conclusion, while it was proposed that epigenetic silencing of the ER receptor might play a role in endocrine resistance in breast cancer, our results in cfDNA did not find support for this. We note that the evaluation of the tumor’s *ESR1* methylation status in cfDNA may be confounded by the presence of a background age-related methylation signal of *ESR1* in WBCs.

## 4. Methods

### 4.1. Clinical Samples

We included plasma samples from patients with ER+/HER2- MBC, selected from three studies. Cohort 1 comprised patients starting treatment with first-line aromatase-inhibitors (CareMore-AI study, trialregister.nl, number NL4884), whereas cohort 2 included patients progressing on any line of endocrine therapy (TAX-ESR1 study on breast cancer, trialregister.nl, number NL7280 and the CareMore-Trastuzumab study on breast cancer, trialregister.nl, number NL4977). The study was performed in accordance with the Declaration of Helsinki. All patients gave written informed consent prior to study procedures. The median age of patients in both cohorts was 66 years (cohort 1 range: 47–88 years; cohort 2 range: 55–70 years). In cohort 1, samples were collected in EDTA tubes and centrifuged within 24 h after collection. Using the same tube type and processing, samples from 20 female healthy donors with a median age of 58 years (range: 55–74 years) were used as a reference. In cohort 2, samples were collected in CellSave tubes and isolated within 96 h after collection. Using the same tube type and processing, samples from 10 female healthy donors with a median age of 61 years (range: 55–70 years) were used as a reference. EDTA (*n* = 20) and CellSave plasma (*n* = 10) processed similarly from age- and gender-matched healthy blood donors (HBD) were used as references. cfDNA was isolated from plasma using the Maxwell^®^ (MX) RSC ccfDNA Plasma Kit (Promega, Madison, WI, USA) in cohort 1 and the QIAamp Circulating Nucleic Acids kit (Qiagen, Venlo, The Netherlands) in cohort 2.

### 4.2. ESR1 Mutations

For all samples, *ESR1* mutations were detected using the Oncomine™ Breast Cancer panel containing 10 breast cancer specific genes and including molecular barcoding to establish a limit of detection (LOD) as low as 0.1% (Life Technologies, Carlsbad, CA, USA). For all samples, 10 ng DNA input was used for library preparation. Samples with a sequencing coverage <500 molecules were excluded from analysis. Additionally, variants were annotated as true variants when they were identified in at least 3 unique molecules.

### 4.3. Culturing Cell Lines and DAC-Treatment

MDA-MB-436 and HCC1187 breast cancer cells (ATCC, HTB-130 & CRL-2322; authenticated by Short Tandem Repeat profiling) were cultured in RPMI1640 glutamax medium (Thermo Fisher Scientific, Waltham, MA, USA) supplemented with 10% heat-inactivated fetal bovine serum, 100 µg/mL penicillin, and 80 µg/mL streptomycin. To analyze restoration of ER mRNA expression following demethylation, MDA-MB-436 cells were treated with 0.1 µM, 0.2 µM, and 0.5 µM 5-aza-2′ deoxycytidine (DAC) (Sigma, St. Louis, MO, USA) and diluted in DMSO daily for 5 days, along with untreated and mock-treated cells. The HCC1187 cell line was excluded from DAC treatment because *ESR1* is unmethylated in this cell line [26] (GSE57342). Genomic DNA or RNA was extracted from cell line pellets using the Nucleospin Tissue kit (Macherey-Nagel, Düren, Germany) or Trizol (Tebu-Bio, Heerhugowaard, The Netherlands), respectively, according to the manufacturer’s instructions.

### 4.4. ESR1-Targeted Quantitative Methylation-Specific PCR (qMSP)

Based on the RefSeq-curated sequence, *ESR1* has 3 promoter regions, referred to as promoter A, B, and C, according to literature (Figure 4) [12]. Of these, only promoter A is located within a CpG island. The methylation status of *ESR1* promotor region A of DAC-treated and control cell lines and patient cfDNA was determined by qMSP analysis after sodium bisulfite conversion (BSC). BSC was performed with the EZ DNA Methylation kit (Zymo Research, Orange, CA, USA) using 10 ng cfDNA for patient samples and with 500 ng genomic DNA for cell line samples following the manufacturer’s instructions. We adapted previously published *ESR1* primers [17,25,27] to amplify the methylated promotor region of *ESR1* in a duplex reaction with the β-actin (*ACTB*) reference gene. Details of the primers and probes used are given in Appendix A. The sensitivity and efficiency of our duplex assay were determined with the MDA-MB-436 cell-line DNA, which is methylated at promoter A of *ESR1*, in serial dilution using unmethylated HCC1187 DNA after whole genome amplification. Our duplex assay detects methylated *ESR1* down to 0.1% in a background of unmethylated DNA (Appendix A) and down to 0.1 ng of total input of methylated DNA (Appendix A). qMSP reactions were carried out in a final reaction volume of 25 µL, containing 300 nM of each primer, 125 nM of each probe, 12.5 µL 2x EpiTect MethyLight Master Mix (w/o ROX, Qiagen, Venlo, The Netherlands), and 1 µL 50x diluted ROX Dye Solution. qPCR reactions were performed on the Mx3000P and Mx3005P QPCR Systems (Stratagene, La Jolla, CA, USA). Samples with an *ACTB*-Ct value above 32 were excluded for data analysis. Methylation values of *ESR1* were normalized to *ACTB* using the comparative Ct method (2−ΔCT). Genomic DNA from methylated, MDA-MB-436 cells was used as positive control, whereas whole-genome-amplified HCC1187 DNA served as unmethylated control DNA.

### 4.5. Reverse-Transcriptase Quantitative PCR (RT-PCR)

Two μg of the isolated RNA was converted to cDNA using the Thermo Scientific RevertAid H Minus First Strand cDNA Synthesis Kit (Fermentas, Thermo Scientific, Waltham, MA, USA), followed by an RNase H step (Ambion, Life Technologies, Carlsbad, CA, USA) to digest any remaining RNA. Quantitative real-time PCR targeting exon 4 of ESR1 [28] was performed in triplicate with the Mx3000P QPCR machine (Agilent Technologies, Santa Clara, CA, USA) using ABgene Absolute Universal or Absolute SYBR Green with ROX PCR reaction mixtures (Thermo Scientific, Waltham, MA, USA) according the manufacturer’s instructions.

### 4.6. Statistical Analysis

cfDNA from healthy donors (EDTA *n* = 10, CellSave *n* = 20) was used to determine a cut-off for *ESR1* methylation positivity, which was defined as the upper limit of the 95% confidence interval. Cut-offs were separately determined for EDTA and CellSave tubes. Statistical tests used are specified throughout the Results section. *p* values < 0.05 were considered statistically significant. Statistical analysis was performed using SPSS Statistics, version 25.0 (ICM Co., Armonk, NY, USA).

## Figures and Tables

**Figure 1 ijms-23-05631-f001:**
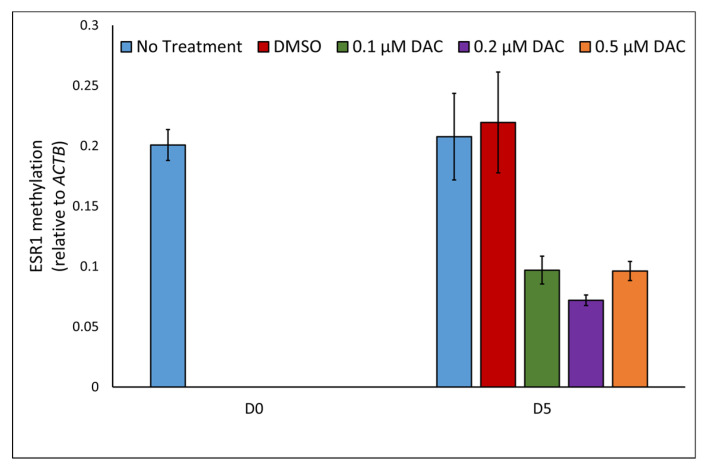
Restoration of *ESR1*/ER after DAC treatment. (**Top**): Methylation decrease in *ESR1* promotor region A is observed in MDA-MB-436 after DAC treatment by qMSP using the 2−ΔCt method, compared to the *ACTB* housekeeping gene. (**Bottom**): Increased ER expression, measured by targeting *ESR1* exon 4 mRNA after cDNA synthesis, is observed in MDA-MB-436 after DAC treatment by qPCR using the 2−ΔCt method compared to the average of the *PBGD* and *HPRT1* housekeeping genes. Error bars represent the standard deviation. * *ESR1* expression levels were multiplied 250× for scale unification.

**Figure 2 ijms-23-05631-f002:**
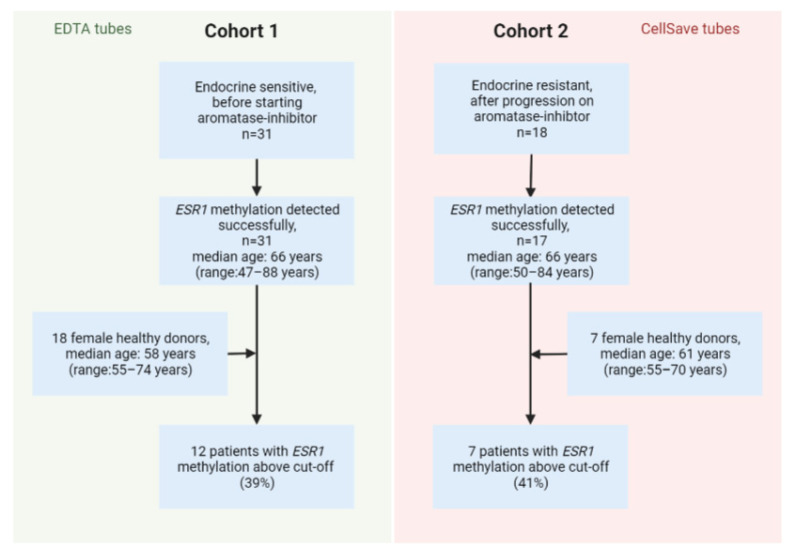
Flow chart of patient samples and healthy donor samples that were included in this study.

**Figure 3 ijms-23-05631-f003:**
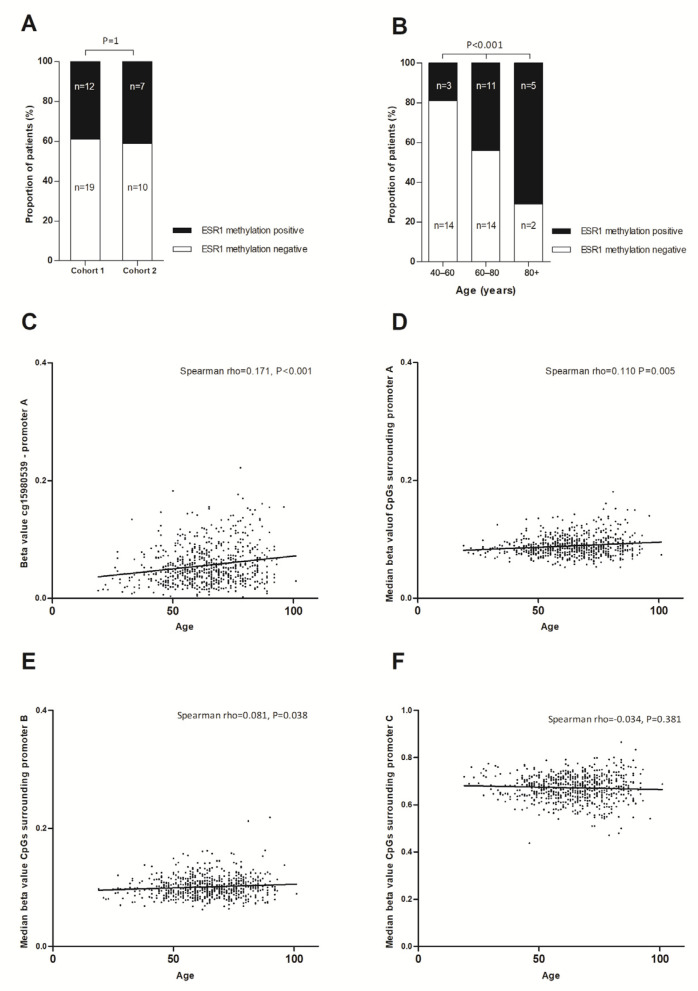
(**A**,**B**). *ESR1* methylation in cfDNA from patients from cohorts A and B. (**A**). Prevalence of *ESR1* methylation in cfDNA is not increased in endocrine-resistant patients; Fisher’s exact test (**B**). Prevalence of *ESR1* methylation in cfDNA increased with age; chi-square test for trend (**C**–**F**). *ESR1* methylation in cfDNA from healthy subjects, and association with age. Data were publicly available and downloaded from Hannum et al. Mol Cell 2013. (**C**). Beta value of cg15980539, targeted by our *ESR1* methylation assay (**D**). Median beta value of CpG dinucleotides located at promoter A (**E**). Median beta value of CpG dinucleotides located at promoter B (**F**). Median beta value of CpG dinucleotides located at promoter C.

**Figure 4 ijms-23-05631-f004:**
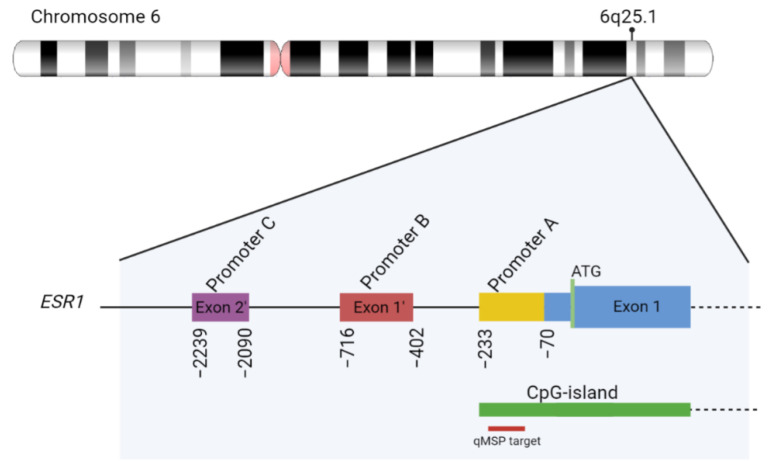
*ESR1* exon 1 and positions of *ESR1* promoter regions, based on the RefSeq curated *ESR1* sequence. The genomic coordinates are relative to the *ESR1* startcodon located at chr6:151,807,913 (hg38 reference genome). The red line indicates the location of the qMSP that was used to evaluate the *ESR1* methylation status in cohorts 1 and 2.

## Data Availability

Sequences of the primers and probes that were used are provided in the Appendix A. Methylation data from white blood cells were publicly available and downloaded from NCBI’s Gene Expression Omnibus, using the following accession number: GSE40279.

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
