# Peer review of "ESR1 Methylation Measured in Cell-Free DNA to Evaluate Endocrine Resistance in Metastatic Breast Cancer Patients"

_ijms, 2022, doi:10.3390/ijms23105631_

Round 1

Reviewer 1 Report

This paper is a study confirming the ESR1 DNA methylation pattern related to endocrine resistance in cfDNA.

Since cfDNA is a non-invasive method and DNA methylation is an important factor in diagnosis and prognosis, it is a necessary study.
However, the level of results is very insufficient to be published in IJMS. It is suggested to submit revisions to other journals that match the level of this paper.

  1. Figure 1 is properly visualized. However, the exact genomic location of the ESR1 gene promoter and the genome assembly version (e.g. hg38) should be presented.
  2. Overall Manuscript: Write words of Latin origin, such as in vitro, in italics.
  3. In Figure 2, unify the y-axis scale and add a label. Significant digits of the decimal point must also be unified. Maybe not "beta value"? And the overall resolution of the figure is low.
  4. Strategies that confirmed the relationship between ESR1 methylation and age using Hannum 2013 data are good in this study. For each plot of C-F in Figure 3, please provide the accession number (e.g. cg15980539) for each location, and suggest the genomic location and gene symbol. Using the IlluminaHumanMethylationEPICanno.ilm10b2.hg19 R package, you can obtain genomic information for each accession number.
  5. Line 154 "en" maybe typo error?
  6. Line 242 "RNAse" -> "RNase"
  7. Line 248: Provide donor information, such as age, gender, etc. and provided as a table.

Reviewer 2 Report

General comments

This manuscript entitled ESR1 methylation measured in cell-free DNA to evaluate endocrine resistance in metastatic breast cancer patients aims to evaluate ESR1 methylation potential as a minimally-invasive biomarker in metastatic breast cancer. The authors investigated ESR1 methylation in cell-free DNA and its association with endocrine resistance in MBC patients.

The manuscript is well written, especially for non-English natives. In my opinion, some points of the paper need to be revised. The major limitation of the study seems to be the sample size.

Abstract

  1. Please introduce the abbreviation cfDNA after cell-free DNA in the abstract. Take care of the ESR1 italicization (the same in the whole manuscript, e.g., page 2, line 52 ACTB).

Introduction

The Introduction section explains the scientific background and rationale for the reported investigation. This manuscript section is well written.

  1. As the NF1 gene alias appears in the MB only ones, please write the full name of the gene.

Results

  1. IJMS requires the non-standard arrangement of chapters; thus, some flaws appear in the manuscript that should be corrected.
  2. The first paragraph is more Materials and methods instead Result section of the investigation described.
  3. Subheading 2.3. describes Cohort1 and Cohort2 methylation prevalence in cfDNA. Please explain abbr. AI (aromatase inhibitors?).
  4. How were the age ranges established? Additionally, the 40-60yrs group consists of only three patients. Is the sample size in this group not too small to draw a significant conclusion about age-related increasing methylation with age (line 88)?

Discussion

  1. The discussion is closely related to the aim and results of the study. It is well written and discusses the outcomes based on the literature.
  2. Please add the information that “epigenetic silencing of the ER receptor might play a role in endocrine resistance in breast cancer … “ in the last paragraph of the discussion.

Methods

  1. The Methods should be extended and describe patients in detail (e.g., in the results section, there is an age-related trend mentioned, but in samples and cohort characteristics, there is a lack of data).
  2. It would be more transparent for the reader to see at least a table or better graph/workflow chart of the samples and controls, differences in cohorts, and participants number (It is hard to check that cohort 1 consists of 31 participants and cohort 2 to of 18).
  3. Lines 251-252 – Sentence “P values <0.05 were considered statistically significant” is duplicated.

Figures

  1. Please add the Y-axis description in Figure 2. Explain all symbols used in the figure and figure legend; it should be self-explanatory.
  2. Line 123-124 – “This association with age was seen for all CpGs in the region of promoter A (Figure 3D) but not for CpGs in the region of promoter B (Figure 3E).” Maybe the correlation for promoter B was not spectacular (I would say very weak) but still significant (p=0.038). Thus the sentence is not correct.

Reviewer 3 Report

In this manuscript by Bos et al., the authors have investigated if DNA methylation of Estrogen Receptor 1 (ESR1) gene promoters in cell free DNA samples can be linked to endocrine resistance. Though the sample size is low, they show evidence that ESR1 promoter methylation does not corelate with endocrine resistance. This is an important piece of evidence for breast cancer prognosis studies. I have some suggestions for improving the manuscript before final publication.

  1. The authors should provide small introduction to the physiological role ESR1 and its complex gene regulation before it’s link to cancer. It will be helpful for broader audience to understand the importance of studying ESR1
  2. In Page 2, lane 59, the authors should mention 5-aza-2′-deoxycytidine (DAC) instead of DAC.
  3. In the method section, the authors mention culturing MDA-MB-436 and HCC1187 breast cancer cells. However, the data for DAC is shown for only MDA-MB-436 and they have used HCC1187 as unmethylated control. They should state rational for doing so and cite the literature for differential methylation status of ESR1 promoter between cell lines.
  4. In figure1, the ESR1 methylation analysis data has error bars. Are these changes statistically significant? Why there is no error bars shown for ESR1 expression? Is this reproducible?
  5. In Figure1 legend, the authors should mention ESR1 exon4, instead of ERX4.

Reviewer 4 Report

Dear Prof. Wilting,

I read your article entitled: ESR1 methylation measured in cell-free DNA to evaluate endocrine resistance in metastatic breast cancer patients. 

I have following few concerns about the article.

  1. the introduction and literature is too short. the background data is highly needed to refer the significance this proposed model. specially when it referred the reported  mechanisms underlying endocrine resistance in first paragraph, the references are not sufficient.
  2. in the second paragraphs it is mentioned that fraction is tumor derived DNA (ctDNA) in blood, which can be effectively used to monitor treatment response....it is worthwhile to explain the ration of this fraction, please add reference.
  3. the Acronyms used in the study, should always be described when they come first time in the text, e.g. CAD, CpG..though they were explained in the text in material and method part. 
  4. there are few words, which seemed to be condensed, they should carefully be checked.

hopefully this is useful to re submit the article after these corrections.

Best regards,

Saima

Round 2

Reviewer 1 Report

This manuscript has been appropriately revised according to the reviewer's comments.

Author Response

We thank the reviewer for his/her positive feedback.